# SeedLM: Compressing LLM Weights into Seeds of Pseudo-Random Generators

**Rasoul Shafipour**[1], **David Harrison**[1], **Maxwell Horton**[1], **Jeff Marker**[1], **Houman Bedayat**[1],

**Sachin Mehta**[2], **Mohammad Rastegari**[3], **Mahyar Najibi**[1], **Saman Naderiparizi**[1]

[1]Apple [2]University of Washington [3]Meta AI

## Abstract

Large Language Models (LLMs) have transformed natural language processing, but face significant challenges in widespread deployment due to their high runtime cost. In this paper, we introduce SeedLM, a novel post-training compression method that uses seeds of a pseudo-random generator to encode and compress model weights. Specifically, for each block of weights, we find a seed that is fed into a Linear Feedback Shift Register (LFSR) during inference to efficiently generate a random matrix. This matrix is then linearly combined with compressed coefficients to reconstruct the weight block. SeedLM reduces memory access and leverages idle compute cycles during inference, effectively speeding up memory-bound tasks by trading compute for fewer memory accesses. Unlike state-of-the-art methods that rely on calibration data, our approach is data-free and generalizes well across diverse tasks. Our experiments with Llama3 70B, which is particularly challenging, show zero-shot accuracy retention at 4- and 3-bit compression to be on par with or better than state-of-the-art methods, while maintaining performance comparable to FP16 baselines. Additionally, FPGA-based tests demonstrate that 4-bit SeedLM, as model size increases, approaches a 4x speed-up over an FP16 Llama 2/3 baseline.

## 1 Introduction

Large Language Models (LLMs) have demonstrated impressive performance across numerous benchmarks (Achiam et al., 2023; Touvron et al., 2023). However, the practical deployment of these models often encounters limitations due to substantial memory transfer requirements. This issue is especially pronounced during autoregressive generation, which is primarily memory-bound and takes the majority of the inference time (Lee et al., 2024). In contrast, operations like 8-bit integer multiplication performed at 45nm 0.9V are demonstrated to be over 200x more energy-efficient than reading the same 8 bits from DRAM (Horowitz, 2014). In this paper, we explore the following question: Can we trade increased compute for a reasonable reduction in memory accesses? A positive answer here not only transforms energy-intensive memory access operations into more energy-efficient compute operations but also alleviates the memory bandwidth limitations that pose a significant bottleneck during LLM inference.

Post-training weight compression is an effective method to reduce the size of pretrained LLMs, making them suitable for on-device execution or reducing power consumption through fewer memory reads. Current state-of-the-art techniques for compressing weights typically require calibration data and involve meticulously adjusting the weights to ensure that the learned knowledge is retained.

We introduce SeedLM, a simple yet effective compression technique which can compress weights to 3-4 bits with minimal accuracy loss. SeedLM is an innovative method for compressing the weights of LLMs by projecting weight blocks into pseudo-random projection basis sets. By finding the optimal seeds to generate these pseudo-random projections per weight block, SeedLM ensures a low compression error and consequently maintains the accuracy of the original model. Our approach

---

Corresponding author: {rshafipour}@apple.com. All work was conducted at Apple.

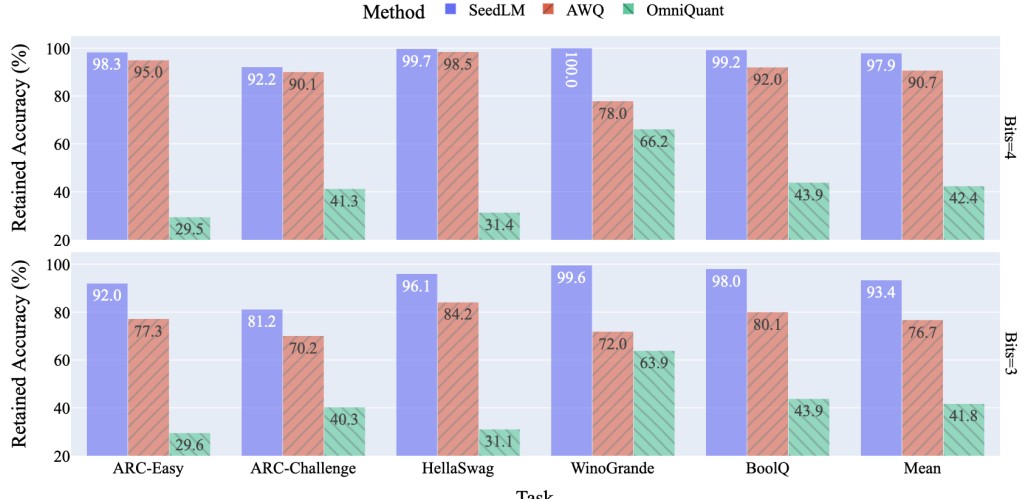

Figure 1: Retained zero-shot accuracy across a variety of tasks and compression methods, compared to the standard Llama 3 70B model. The top row shows data for 4-bit compression, while the bottom row shows data for 3-bit compression. We compare the performance of SeedLM, AWQ, and OmniQuant across the ARC-Easy, ARC-Challenge, HellaSwag, WinoGrande, and BoolQ tasks. While being completely data-free, SeedLM outperforms state-of-the-art weight quantization methods that rely on a calibration dataset.

only requires storing the seed and few projection coefficients instead of all the weight values to reconstruct high dimensional weight blocks. As a result, SeedLM significantly reduces the memory footprint required for operating large-scale models during inference. To generate pseudo-random matrix blocks given a seed, we leverage Linear Feedback Shift Register (LFSR) hardware blocks that are widely used in applications such as cryptography, communication, error detection, etc (Gaitonde & Ramabadran, 1988; Zeng et al., 2013; Xiang et al., 2016). LFSRs can be efficiently implemented in the silicon with minimal energy and area footprint.

Figure 1 shows the Retained Accuracy (%), which is the ratio of the compressed model's accuracy to the full-precision FP16 model's accuracy, on the Llama 3 70B model. As shown, SeedLM retains approximately 97.9% of zero-shot accuracy across various tasks in a data-free setting, using 4 bits per weight element (see Section 4.1 for more details). It also consistently outperforms state-of-the-art 3-bit and 4-bit compression techniques that rely on calibration data. To the best of our knowledge, this is the first time nearly identical accuracy has been achieved with 4-bit compression on LLMs without data, using a deterministic offline algorithm.

The summary of our contributions is as follows:

- For the first time, we demonstrate how to leverage LFSR hardware blocks to trade increased compute with lower memory accesses.
- We show the first instance of achieving nearly identical accuracy with 4-bit quantization without data, using a deterministic offline algorithm.
- We demonstrate an effective solution to find the optimized seed for the LFSR modules while maximizing compression ratio.
- We prototype SeedLM on an FPGA (Field-Programmable Gate Array) and demonstrate its efficacy in reducing the inference latency with custom hardware.

## 2 RELATED WORK

Significant research has been conducted in model compression for LLMs, a critical approach for reducing both the memory footprint and computational demands of these models. In this section, we highlight some of the most relevant techniques from prior work.

**Compression With Random Basis:** Recent works have demonstrated that neural networks can be decomposed into random number generator seeds and weight coefficients. In PRANC (Nooralinejad et al., 2022), full networks are compressed by orders of magnitude to improve storage and transmission efficiency. LoRA (Hu et al., 2021) compresses the weights by injecting trainable rank decomposition matrices into each layer of the network. NOLA (Koohpayegani et al., 2023) builds upon LoRA by compressing the low-rank matrices through a linear combination of random basis vectors, further reducing memory and computational overhead.

Our work (SeedLM) is conceptually similar in that we compress networks using random basis. These other models rely on much larger basis ranks applied globally rather than per-block. As a result, these methods need far more operations per parameter to preserve accuracy and are not computationally feasible at inference for LLMs.

**Data-Free Post-Training Compression:** A few previous works have explored data-free post-training compression (Nagel et al., 2019; Horton et al., 2020; Nunez et al., 2023). Such works are capable of producing a compressed model after training without the need for calibration data. They usually apply quantization or pruning techniques to obtain a smaller model. Similarly, SeedLM does not require any data for model compression. This is in contrast to most recent works on LLM compression, which require calibration data.

There are more computationally expensive methods for data-free compression that involve generating data from a teacher model and performing distillation (Lopes et al., 2017; Gou et al., 2020). These techniques are applied to LLMs as demonstrated in Liu et al. (2023).

**Post-Training Compression with Calibration Data:** An early example of post-training quantization with calibration data is found in Nagel et al. (2020), where activation statistics are used to decide whether to round quantized values up or down. The cost of post-training quantization is a small fraction of the total training cost.

Recently, calibration data has been leveraged for post-training compression in LLMs. AWQ (Lin et al., 2024) rescales salient weights before compression using activation statistics. In QuIP# (Tseng et al., 2024) and GPTQ (Frantar et al., 2022), Hessian analysis of calibration data helps make rounding decisions during quantization. SpQR (Dettmers et al., 2023) retains outliers during quantization to preserve accuracy. OmniQuant (Shao et al., 2023) employs weight clipping and other transformations to maintain accuracy. Additive Quantization (Egiazarian et al., 2024) learns a codebook for performing additive quantization. In our study, we used AWQ and OmniQuant as our main baselines because they avoid costly training while delivering strong results.

**Training-Aware Compression:** Compressing the model during the training process has the disadvantage of fixing the compression method and parameters beforehand, but usually offers better accuracy. An overview of quantization-aware training is provided in Jacob et al. (2017); Nagel et al. (2022); Xi et al. (2023), while pruning techniques are discussed in Alizadeh-Vahid et al. (2023); Sun et al. (2023); Kusupati et al. (2020); LeCun et al. (1989). In Rouhani et al. (2023), stable training and post-training quantization are demonstrated using hardware-friendly 4-8 bit weights, activations, and gradients with minimal accuracy loss by utilizing micro-scaled data formats.

In this work, we focus on post-training weight compression with SeedLM, and in the following sections, we outline our methodology and present the results.

## 3 METHODOLOGY

In this section, we introduce SeedLM, our method for compressing the weights of LLMs by using seeds from a pseudo-random generator. Initially, each weight matrix is segmented into blocks of $C$ contiguous elements. Representing each block as a vector $\mathbf{w} \in \mathbb{R}^C$, we approximate it as a linear combination of columns from a matrix $\mathbf{U} \in \mathbb{R}^{C \times P}$. This matrix $\mathbf{U}$ is constructed using a pseudo-random generator given a seed specifically selected to generate a subspace that most effectively reconstructs $\mathbf{w}$ linearly.

Figure 2 illustrates this setup. Our primary goal is to find the optimal seed, $s$, and coefficient vector, $\mathbf{t} \in \mathbb{R}^P$, that minimize the reconstruction error between the original and the approximated weights. For this reconstruction, only the seed and the coefficients are stored. In the following subsection, we first outline a mechanism to efficiently generate $\mathbf{U}$ using a $K$-bit seed in our Linear Feedback Shift Register (LFSR) framework. We will then discuss the methodologies used to determine $s$ and $\mathbf{t}$.

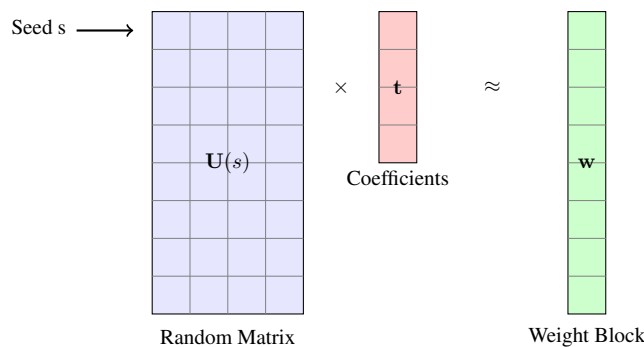

Figure 2: Compression of weights using pseudo-random generated matrices.

## 3.1 LINEAR FEEDBACK SHIFT REGISTER (LFSR)

A Linear Feedback Shift Register (LFSR) is a simple yet effective type of shift register, ideal for generating pseudo-random binary sequences. The primary advantages of LFSRs in hardware include cost-effectiveness and minimal resource consumption due to their straightforward implementation with basic flip-flops and XOR gates. This simplicity facilitates rapid and efficient sequence generation, which is integral to our compression technique.

An LFSR operation can be characterized by its length $K$ (which determines the number of bits in its shift register) and its feedback polynomial. To generate next pseudo-random number in the sequence, each bit in the register is first shifted to the next position. Then, the new bit entering the register is calculated as a linear combination of certain bits of the current state as specified by the feedback polynomial, typically implemented by XOR operations. Mathematically, the new bit $x_{n+1}$ generated by the LFSR can be expressed as:

$$x_{n+1} = \sum_{i=0}^{K-1} \alpha_i \cdot x_{n+i-K+1} \mod 2,$$

where $K \geq 2$ and $\alpha_0, \ldots, \alpha_K$ are the binary coefficients that define the feedback polynomial, with each $\alpha_j$ determining whether the bit $x_j$ is selected or not.

The state transition in the LFSR can be described as follows: if the current state is represented by the bits $x_n, x_{n-1}, \ldots, x_{n-K+1}$, then after the shift, the new state will be $x_{n+1}, x_n, \ldots, x_{n-K+2}$. This transition reflects the shift of every bit to the right by one position, with the new bit $x_{n+1}$ entering at the leftmost position. Given its finite state nature, an LFSR will eventually enter a repeating cycle, suggesting an asymptotic uniform distribution over this cycle. An LFSR can cycle through at most $2^K - 1$ states, excluding the all-zero state.

A key goal when designing an LFSR is to guarantee a maximal-length sequence. This ensures that the LFSR will produce the longest possible sequence of non-repeating states before repeating. Intuitively, this means the LFSR will cycle through every possible state (except the all-zero state), maximizing the number of distinct pseudo-random values generated. To achieve this maximal-length property, the feedback polynomial must be primitive over the Galois field GF(2). In simple terms, a primitive polynomial ensures that the LFSR explores all $2^K - 1$ states without prematurely entering a repeating cycle.

For our experiments, this means a fixed set of coefficients $\{\alpha_j : 0 \leq j \leq K-1\}$ that is hard-wired, ensuring maximal length if and only if it avoids the all-zero state, in which it stays in zero. A summary of the algorithmic implementation can be found in Appendix A.3. Refer to Section A.1 for the indexed $j$ coefficients used for each $K$ where $\alpha_j$ equals one; all other coefficients are zero. For a comprehensive understanding of LFSRs and their properties, see (Bhattacharjee & Das, 2022).

To optimize the efficiency of generating random matrices through an LFSR, for a fixed length $K$ and a set of coefficients $\{\alpha_j\}$, we cache all the $2^K - 1$ states that sequentially follow each other – each state uniquely determined by its preceding state along with $K$ and $\{\alpha_j\}$. This cache allows

us to extract an arbitrarily sized random matrix from the sequence given a random seed $s$. We can cycle through these states to generate matrices of any desired size. For an illustration, refer to Section A.2. With $K = 16$ and a maximal length LFSR, all states will occupy approximately $(2^{16} - 1) \times 2$ Bytes $\approx 130$KB of memory, which is negligible. This setup ensures a highly efficient and scalable mechanism for generating the necessary random matrices for our compression technique. $K$ is a hyper-parameter of our method which we will elaborate on in Section 3.4.

## 3.2 WEIGHT COMPRESSION USING PSEUDO-RANDOM GENERATORS

Building on the foundations laid by the LFSR mechanisms, our methodology seeks to represent a block of data $\mathbf{w} \in \mathbb{R}^C$ using the decomposition $\mathbf{U}(s)\mathbf{t}$. Here, $\mathbf{U}(s) \in \mathbb{R}^{C \times P}$ is a random matrix derived from a $K$-bit LFSR. During inference, $s$ and $\mathbf{t}$, which require fewer bits when considering all the bits in a block compared to the original weights, are retrieved from memory. This reduces the memory footprint and accelerates memory-bound generation tasks. The weights are reconstructed on-the-fly using $s$ and $\mathbf{t}$, and these reconstructed weights are then used to compute intermediate activations and, ultimately, the outputs. As the output of the LFSR is in the range of $[1, 2^K - 1]$, which are unsigned integers, we normalize them to ensure they fall within the range of $[-1, 1]$, enhancing their expressiveness. More specifically, to construct $\mathbf{U}(s)$, we first generate a matrix $\mathbf{V}(s)$ using an LFSR seeded by $s$ as illustrated in Figure 4. This matrix initially contains integers and undergoes the following normalization to ensure that its components lie between $[-1, 1]$:

$$\mathbf{U}(s) = \frac{1}{2^{K-1} - 1} \left( \mathbf{V}(s) - 2^{K-1} \mathbf{1} \right),$$

where $\mathbf{1} \in \mathbb{R}^{C \times P}$ represents a matrix of all ones, and K is the LFSR bit length. To determine the optimal seed and coefficients, we solve the following optimization problem:

$$\underset{s, \mathbf{t}}{\text{minimize}} \quad \|\mathbf{w} - \mathbf{U}(s)\mathbf{t}\|_2^2, \quad \text{subject to:} \quad s \in \{1, \ldots, 2^K - 1\}, \quad \text{and} \quad \mathbf{t} \in \mathcal{Q}, \tag{1}$$

where $\|\cdot\|$ denotes the Euclidean norm, and $\mathcal{Q}$ represents the set of valid quantized values. The objective is to identify an optimal seed $s^*$ and coefficients $\mathbf{t}^*$ that most effectively approximate $\mathbf{w}$. This problem is NP-hard due to the discrete nature of the seed selection and the quantization of coefficients since it involves searching through a combinatorially large set of possibilities. Next, we describe the design choices and heuristics used to solve the problem.

The quantization scheme for the vector $\mathbf{t}$ plays a critical role in balancing reconstruction accuracy with bit efficiency, adhering to our bit budget constraints. We represent each element of $\mathbf{t}$ as a 4-bit 2's complement integer, paired with a shared 4-bit exponent. This configuration allows us to capture a broad dynamic range of values within the interval $[-8 \times 2^{-8}, 7 \times 2^7]$. The shared 4-bit exponent extends the dynamic range, enabling representation across several orders of magnitude. We have specifically chosen for the exponent to be a power of two, which is hardware-friendly and can be efficiently implemented using simple shift operations in digital circuits; see (Darvish Rouhani et al., 2020). By combining the 4-bit integer with the shared exponent, each quantized element is expressed as $t_i = q_i \times 2^e$, where $q_i$ is the 4-bit 2's complement integer and $e$ is the shared exponent. The shared exponent $e$ is selected as:

$$e = \max_i \lfloor \log_2 (|t_i|) \rfloor,$$

where $|t_i|$ is the absolute value of each element of $\mathbf{t}$. After determining $e$, each $t_i$ is scaled by dividing it by $2^e$, and then quantized to a 4-bit 2's complement integer to derive $q_i$.

## 3.3 APPROXIMATION APPROACH

Looking back at the optimization problem in Eq. 1, while the unconstrained case admits a closed-form solution given by $\mathbf{U}(s)^\dagger \mathbf{w}$, where $\mathbf{U}(s)^\dagger$ denotes the Moore-Penrose pseudo-inverse of $\mathbf{U}(s)$, our discrete constraints convert it to an NP-hard optimization problem. Hence, to solve Eq. 1, we employ an approximate heuristic approach that involves the following steps:

1. Generate $N = 2^K - 1$ random matrices $\{\mathbf{U}(s_1), \mathbf{U}(s_2), \ldots, \mathbf{U}(s_N)\}$, each of size $C \times P$, with an LFSR of length $K$.

2. For each matrix $\mathbf{U}(s_j)$, project the vector $\mathbf{w}$ onto the subspace spanned by $\mathbf{U}(s_j)$:

$$\mathbf{t}_j = \mathbf{U}(s_j)^{\dagger}\mathbf{w}.$$

3. Quantize $\mathbf{t}_j$ to obtain the vector $\hat{\mathbf{t}}_j$ using 4-bit integers and a 4-bit shared exponent $e_j$.

4. Compute the reconstruction error for each pair $(\mathbf{U}(s_j), \hat{\mathbf{t}}_j)$ as follows:

$$\epsilon_j = \|\mathbf{w} - \mathbf{U}(s_j)\hat{\mathbf{t}}_j\|_2^2.$$

5. Select the pair $(s^*, \mathbf{t}^*)$ that minimizes the reconstruction error:

$$(s^*, \mathbf{t}^*) = \arg\min_{s_j, \hat{\mathbf{t}}_j} \epsilon_j. \tag{2}$$

Our heuristic algorithm leverages randomness to explore multiple subspaces and selects the one that best approximates $\mathbf{w}$ under the given constraints. In summary, we apply the above heuristics across all weight blocks in parallel to find the seeds and coefficients that minimize the reconstruction error. To enhance computational efficiency, we precompute and cache the pseudo-inverse matrices for all seeds and perform steps 2–5 in parallel across all blocks. The complete algorithm is provided in Appendix A.5.

## 3.4 DESIGN SPACE EXPLORATION

The minimum reconstruction error obtained from Eq. 2 depends on the block size $C$, the latent dimension $P$, and the LFSR length $K$. Here, we explore how we select the optimal configuration for an $M$-bit compression. First, let's examine the total number of bits required to store a SeedLM block of $C$ elements, which consists of the following:

- $K$ bits to index the selected seed $s^*$ to generate matrix $\mathbf{U}(s^*)$ among $N = 2^k - 1$ candidates.
- 4 bits to store the shared exponent $e$.
- $4P$ bits to store the quantized vector $\mathbf{t}^*$ ($P$ elements each requiring 4 bits).

So, the effective bit per element is a function of hyper-parameters $K$, $C$, and $P$. In particular, for an $M$-bit compression, we have the bit budget per element as

$$M = \frac{K + 4 + 4P}{C}. \tag{3}$$

To determine the optimal configuration for a given bit budget per element, $M$, we evaluate the reconstruction accuracy of our method in the search space. Specifically, we explore how a standard normal Gaussian vector $\mathbf{w}$ can be approximated using any combination of valid hyperparameters given the bit budget $M$. While assuming a Gaussian distribution may have its limitations, it has proven effective within our design space and aligns well with real-world benchmarks. Our objective is to find appropriate values for block size $C$, latent dimension $P$, and LFSR length $K$, such that the reconstruction error is minimized when the optimal seed is selected. More specifically, let $s^*_{C,P,K}$ and $\mathbf{t}^*_{C,P,K}$ denote the solutions obtained from Eq. 2. For an $M$-bit compression, we solve the following optimization problem:

$$\mathbb{E}[\epsilon_{\min}] := \min_{C,P,K} \quad \mathbb{E}\big[\|\mathbf{w} - \mathbf{U}(s^*_{C,P,K})\mathbf{t}^*_{C,P,K}\|_2^2\big],$$
$$\text{subject to:} \quad MC = K + 4 + 4P \quad \text{and} \quad C, K, P \in \mathcal{Z}^+, \tag{4}$$

where $\mathcal{Z}^+$ represents the set of all positive integers. Since the optimization problem in equation 4 is not analytically tractable, we numerically solved it by conducting a grid search over $C$, $P$, and $K$ constrained to positive integers. Understanding the trade-offs among $C$, $P$, and $K$ is important for optimizing the approximation within a given bit budget. Each of these parameters influences the overall performance and contributes to minimizing the reconstruction error.

One critical trade-off is between the LFSR length $K$ and the latent dimension $P$. Increasing $K$ reduces the expected minimum error $\mathbb{E}[\epsilon_{\min}]$ by providing more opportunities to find a better projection. However, this comes at a cost: as $K$ increases, the number of required bits grows, reducing

the available bit budget for $P$. The objective is to find an optimal $K$ that effectively lowers $\mathbb{E}[\epsilon_{\min}]$ without overly constraining $P$, as that could lead to a significant increase in the overall error $\mathbb{E}[\epsilon_{\min}]$. Similarly, increasing $P$ helps capture more of the energy of the vector $\mathbf{w}$, thus reducing $\mathbb{E}[\epsilon_{\min}]$. However, this also requires more bits, which may limit the value of $K$. The key is to strike a balance where enough energy is captured without overly sacrificing the exploration of better projections through $K$. Finally, increasing $C$ expands the total bit budget, allowing for larger values of both $P$ and $K$. However, this also increases the potential for higher error, as expanding the space may dilute the precision of projections.

Overall, $C$, $P$, and $K$ are interdependent, and optimizing these parameters requires a careful numerical exploration of different combinations. Based on this analysis, we selected the configurations shown in Table 1 for $M=3$ and $M=4$, which are used in the experiments reported next.

Table 1: Selected configurations of $C$, $P$, and $K$ for $M = 3$ and $M = 4$.

| bits per element $M$ | Block size $C$ | Latent Dimension $P$ | LFSR Seed Bits $K$ |
|---|---|---|---|
| 3 | 12 | 4 | 16 |
| 4 | 8 | 3 | 16 |

## 4  EXPERIMENTS

In this section, we apply the heuristics across all weight blocks of pretrained LLMs in parallel to find the seeds and coefficients, as described in Eq. 2, that minimize the reconstruction error. Using the configurations from Table 1, we evaluate our compression methods in terms of accuracy and performance. Our experiments focus on Llama 2 and Llama 3 models (Touvron et al., 2023), and unlike other methods, SeedLM does not require fine-tuning or calibration data while still achieving competitive results. We assess model quality using perplexity and accuracy, followed by performance analysis through FPGA-based matrix multiplication with low-level LFSR generation. This highlights the cost and performance benefits of SeedLM, especially in hardware-constrained environments.

### 4.1  ACCURACY RESULTS

To evaluate the quality of SeedLM, we measure perplexity on the WikiText-2 dataset (Merity et al., 2016) and assess accuracy across various zero-shot tasks using LM Evaluation Harness (Gao et al., 2021)[1]. We compare our method against established compression techniques such as AWQ (Lin et al., 2024) and OmniQuant (Shao et al., 2023), using the official GitHub repositories for each baseline as of September 2024. Additionally, we compare with GPTQ (Frantar et al., 2022) at 4-bits, as summarized in Appendix A.6. A key strength of SeedLM is that it can operate entirely data-free, in contrast to other methods that require calibration data to achieve comparable results. For the baseline methods, we use the default calibration sets from their official repositories. Our experiments involve Llama 2 models (7B, 13B, 70B) and Llama 3 models (8B, 70B), tested with 3-bit and 4-bit representations. In the case of AWQ and OmniQuant, we use 4-bit integers with channel-wise scaling to avoid significantly increasing the bits per element beyond the allocated 3 or 4 bits (since a group size of 128 in these methods adds roughly 0.25 extra bits per parameter). For OmniQuant, we ensure a fair comparison with SeedLM and AWQ by not performing fine-tuning on the quantized models. However, unlike SeedLM, all of them still require per-layer calibration, which relies on calibration data and activations, whereas SeedLM achieves its results without any data dependency.

---

[1]For all compression methods, we use LM Evaluation Harness v0.4.3 and the following task versions: arc-challenge=1.0, arc-easy=1.0, hellaswag=1.0, winogrande=1.0, boolq=2.0.

To evaluate language model performance, we measure perplexity on WikiText-2 using 166 test windows of 2048 tokens each. The results, as shown in Table 2, illustrate a trade-off between compression level and model quality. In some cases, aggressively compressed large models even underperform smaller models with mild compression. SeedLM consistently outperforms state-of-the-art compression techniques, particularly at higher compression levels. Notably, SeedLM achieves these results without the need for any calibration data.

Table 2: WikiText-2 perplexity results for 3- and 4-bit representations of Llama 2 and Llama 3 models with a sequence length of 2048. The notation x-yB refers to the Llama x model with yB parameters (e.g., 2-7B means Llama 2 with 7 billion parameters). Perplexity values above 100 are shown as inf. The best perplexity values are highlighted.

| Method | Bits | 2-7B | 2-13B | 2-70B | 3-8B | 3-70B |
|--------|------|------|-------|-------|------|-------|
| Baseline | 16 | 5.5 | 4.9 | 3.3 | 6.1 | 2.9 |
| SeedLM | 4 | 5.7 | 5.1 | 3.5 | 7.0 | 3.8 |
| OmniQuant | 4 | 6.1 | 5.2 | 3.7 | inf | inf |
| AWQ | 4 | 5.8 | 5.1 | 3.5 | 7.1 | 4.7 |
| SeedLM | 3 | 6.6 | 5.8 | 4.0 | 10.1 | 5.7 |
| OmniQuant | 3 | inf | 10.7 | 7.5 | inf | inf |
| AWQ | 3 | 15.6 | 6.5 | 4.4 | 11.8 | 11.6 |

Next, we show zero-shot accuracy across various tasks. As seen in Table 3, SeedLM performs on par with or better than state-of-the-art methods at the same bit rates. This underscores SeedLM's ability to maintain competitive accuracy without relying on calibration data. Note that Llama 3 is much more sensitive to compression than Llama 2, likely due to its more advanced architecture and significantly larger training dataset. Llama 3 was trained on 15 trillion tokens, around seven times more than Llama 2's 2 trillion tokens, enabling it to capture more detailed language patterns and nuances. This increased complexity and sensitivity to nuance make it less compressible without a noticeable drop in performance. However, as shown in Figure 1, SeedLM still significantly outperforms other state-of-the-art methods on Llama 3 in terms of Retained Accuracy (%), i.e. ratio of the compressed model's accuracy to the full-precision FP16 model's accuracy. This demonstrates that SeedLM is not only effective for such a complex model but is also robust across various tasks by maintaining high accuracy.

## 4.2 Performance Analysis

In this section, we shift our focus from accuracy to performance analysis on hardware. Specifically, we explore how SeedLM can be efficiently implemented on an FPGA. FPGAs are ideal for this task because they allow for highly parallelized computations and can be reconfigured to handle specific workloads, making them well-suited for running compressed models with lower bit rates that are not well supported by conventional GPUs. With an FPGA, the LFSR generation can be supported at a low-level in hardware rather than relying on relatively expensive software implementations.

To evaluate SeedLM on an FPGA, we benchmark matrix-vector multiplication – a core operation in most large language models – both with and without SeedLM's 4-bit parametrization. Figure 3 shows the RTL design block diagram, with the target device being an AMD Virtex7 FPGA XC7V585T-3 (2021).

The implementation utilizes 128 DSP48 slice multipliers in parallel, calculating 128 elements of the activation vector simultaneously. The DDR response interface has a maximum bandwidth of 64 bytes per 200 MHz clock cycle, and the data path is designed to compute at the maximum DDR response throughput. When SeedLM compression is bypassed, the LFSR weight decompression is also skipped.

In the reference design, FP16 weights are read directly from DRAM, bypassing decompression. Due to the DDR interface's 64-byte bus width, a maximum of 32 weight values can be read per cycle, limiting the utilization to only 32 of the 128 MACs. With SeedLM's 4-bit compression, however, 128 weight values can be read per cycle, resulting in a theoretical 4x performance improvement in memory access.

Table 3: Performance comparison across different models and zero-shot tasks for 4-bit and 3-bit configurations.

| Model | Method | Bits | ARC-Easy | ARC-Challenge | HellaSwag | WinoGrande | BoolQ | Mean |
|-------|--------|------|----------|---------------|-----------|------------|-------|------|
| Llama 2 7B | Baseline | 16 | 74.58 | 46.33 | 75.98 | 69.06 | 77.74 | 68.74 |
| | SeedLM | 4 | 73.23 | 44.54 | 74.45 | 68.43 | 77.19 | 67.57 |
| | AWQ | 4 | 70.58 | 43.94 | 74.96 | 68.75 | 78.29 | 67.30 |
| | OmniQuant | 4 | 70.71 | 43.52 | 74.20 | 68.27 | 73.64 | 66.07 |
| | SeedLM | 3 | 69.87 | 41.21 | 70.72 | 66.30 | 74.28 | 64.48 |
| | AWQ | 3 | 53.37 | 33.62 | 56.66 | 61.09 | 57.58 | 52.46 |
| | OmniQuant | 3 | 35.69 | 25.77 | 35.48 | 52.88 | 42.48 | 38.46 |
| Llama 2 13B | Baseline | 16 | 77.44 | 48.98 | 79.38 | 72.22 | 80.55 | 71.71 |
| | SeedLM | 4 | 76.98 | 49.83 | 78.54 | 72.77 | 79.20 | 71.46 |
| | AWQ | 4 | 77.44 | 49.32 | 78.57 | 71.90 | 78.47 | 71.14 |
| | OmniQuant | 4 | 76.18 | 47.95 | 78.10 | 72.14 | 81.77 | 71.23 |
| | SeedLM | 3 | 72.85 | 45.39 | 74.50 | 71.35 | 78.81 | 68.58 |
| | AWQ | 3 | 70.58 | 45.14 | 72.72 | 64.96 | 72.45 | 65.17 |
| | OmniQuant | 3 | 55.85 | 34.47 | 59.54 | 53.04 | 63.39 | 53.26 |
| Llama 2 70B | Baseline | 16 | 80.98 | 57.25 | 83.81 | 77.98 | 83.70 | 76.74 |
| | SeedLM | 4 | 81.14 | 56.40 | 82.97 | 76.72 | 82.26 | 75.90 |
| | AWQ | 4 | 80.98 | 56.66 | 83.24 | 77.19 | 83.27 | 76.27 |
| | OmniQuant | 4 | 79.59 | 55.97 | 82.67 | 76.80 | 83.43 | 75.69 |
| | SeedLM | 3 | 79.00 | 53.84 | 80.51 | 76.80 | 79.02 | 73.83 |
| | AWQ | 3 | 80.26 | 55.80 | 80.50 | 73.01 | 80.00 | 73.91 |
| | OmniQuant | 3 | 63.59 | 39.51 | 68.24 | 62.04 | 65.23 | 59.72 |
| Llama 3 8B | Baseline | 16 | 76.81 | 52.73 | 76.97 | 72.93 | 81.87 | 72.26 |
| | SeedLM | 4 | 76.52 | 49.74 | 76.61 | 72.93 | 80.76 | 71.31 |
| | AWQ | 4 | 74.49 | 51.54 | 78.03 | 73.09 | 80.40 | 71.51 |
| | OmniQuant | 4 | 73.95 | 47.78 | 73.42 | 69.69 | 71.99 | 67.37 |
| | SeedLM | 3 | 67.21 | 41.55 | 68.34 | 69.22 | 67.61 | 62.79 |
| | AWQ | 3 | 64.90 | 40.19 | 68.40 | 65.04 | 74.62 | 62.63 |
| | OmniQuant | 3 | 30.26 | 22.53 | 28.96 | 49.33 | 48.47 | 35.91 |
| Llama 3 70B | Baseline | 16 | 85.23 | 64.33 | 84.07 | 77.66 | 86.27 | 79.51 |
| | SeedLM | 4 | 83.80 | 59.30 | 83.84 | 77.74 | 85.60 | 78.06 |
| | AWQ | 4 | 80.98 | 57.94 | 82.84 | 60.54 | 79.39 | 72.34 |
| | OmniQuant | 4 | 25.13 | 26.54 | 26.36 | 51.38 | 37.83 | 33.45 |
| | SeedLM | 3 | 78.45 | 52.22 | 80.77 | 77.35 | 84.59 | 74.68 |
| | AWQ | 3 | 65.87 | 45.14 | 70.76 | 55.88 | 69.08 | 61.35 |
| | OmniQuant | 3 | 25.21 | 25.94 | 26.15 | 49.64 | 37.83 | 32.95 |

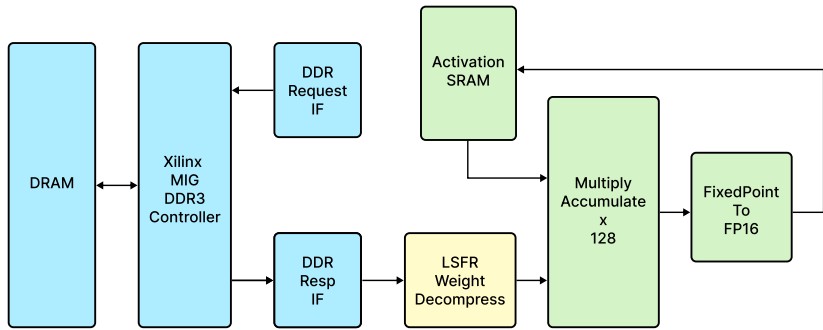

Figure 3: Block diagram of the RTL design.

We use the reference implementation without compression (FP16 for all data) as a baseline to assess both performance and resource utilization on the FPGA. Table 4 provides a detailed comparison of the FPGA resource usage, including LUTs, FFs, BlockRAM, and DSP counts.

- **LUT** – FPGA Fabric Lookup Table, used to create combinational logic.
- **FF** – FPGA Fabric Register (Flip-Flop).
- **BRAM** – FPGA Block RAM (SRAM), each BlockRAM is 36Kb.
- **DSPs** – FPGA DSP48 resources with 18-bit by 18-bit multiplier and accumulator.

In the reference design, only 32 MACs are used due to the input bandwidth limitation of 64 bytes per cycle. The SeedLM design, utilizing 128 MACs per cycle, results in approximately a 4x increase in MAC Block resources, aligning with the expected performance improvement.

Table 4: FPGA resource utilization comparison

| | Total | | MAC Block | | LSFR Decompress | | PreMAC Fix2Float | |
|---|---|---|---|---|---|---|---|---|
| | Reference | SeedLM | Reference | SeedLM | Reference | SeedLM | Reference | SeedLM |
| **LUTs** | 20800 | 120105 | 9902 | 42174 | 0 | 67034 | 0 | 12292 |
| **FFs** | 10164 | 73666 | 3118 | 12594 | 0 | 45407 | 0 | 13448 |
| **BRAMs** | 10.5 | 154.5 | 0 | 0 | 0 | 144 | 0 | 0 |
| **DSPs** | 32 | 128 | 32 | 128 | 0 | 0 | 0 | 0 |

Table 4 reveals that SeedLM increases the total LUT count to 67K and register count to 45.4K, with the fixed-point to FP16 conversion accounting for 12.3K LUTs and 13.4K registers. The design includes 128 fully pipelined fixed-to-FP16 converters to sustain the DDR response data rate. We can note that performing all the compute in fixed-point math could eliminate the need for fixed-point to FP16 conversion, but this optimization is outside the scope of this paper. Table 5 shows the number of cycles required to complete matrix operations of various sizes, measured from the first DDR read to the final write to the activation SRAM. The SeedLM design achieves a 4x throughput compared to the reference design. As matrix size increases, the startup costs become less significant, leading the speedup to approach the theoretical 4x gain. In summary, SeedLM achieves near iso-accuracy at 4-bit compared to FP16, while offering close to a 4x speedup for memory-bound tasks such as generation in LLMs with billions of parameters and beyond.

Table 5: Performance comparison for different matrix sizes

| | $512 \times 512$ | $1024 \times 1024$ | $2048 \times 2048$ |
|---|---|---|---|
| **Reference** | 8593 | 34201 | 136559 |
| **SeedLM** | 2341 | 8723 | 34331 |
| **Speed Up** | 3.67 | 3.92 | 3.98 |

## 5    CONCLUDING REMARKS

In this paper, we presented SeedLM, a post-training compression method that uses pseudo-random generators to efficiently encode and compress model weights. SeedLM offers a data-free approach, avoiding the need for calibration data while retaining competitive accuracy, achieving up to around 98% zero-shot accuracy at 3- and 4-bit quantization levels. We demonstrated the method's performance on both Llama 2 and Llama 3 models, showing that it performs comparably to existing state-of-the-art techniques. Furthermore, our FPGA implementation highlights SeedLM's potential for improved computational efficiency in hardware-constrained environments.

SeedLM primarily focused on 3-bit and 4-bit quantization, applying the same design choices across all blocks for these levels. However, we believe that pushing below the 3-bit level might require different configurations tailored to each block to manage compression noise effectively or leveraging data for additional fine-tuning to enhance results. We leave these directions for future work. Furthermore, as hardware continues to evolve, SeedLM's applicability could expand, with its LFSR-based design offering low-area, low-energy, high-throughput benefits. This work highlights the practicality of such techniques, aiming to inspire hardware innovations that support broader device compatibility, including GPUs.

ACKNOWLEDGMENTS

We are grateful to Dmitry Belenko, Karen Khatamifard, Qingqing Cao, Seyed Mohsen Moosavi Dezfooli, Arsalan Farooq, Moin Nabi, and Devi Krishna for their invaluable feedback and support.

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

## A  APPENDIX

### A.1  COEFFICIENTS USED IN LFSRS

The following table lists the indexed $j$ coefficients used for each $K$, where $\alpha_j$ equals one, with all other coefficients being zero. These specific coefficients correspond to the Linear Feedback Shift Registers (LFSRs) for each register length $K$, with the coefficients indexed starting from 0, representing the tap positions in the shift register. These coefficients are hard-wired into the hardware configuration of the LFSRs used in our experiments. These coefficients define the feedback polynomial for each LFSR, ensuring maximal-length cycles.

Table 6: Indexed $j$ coefficients used in LFSRs for each register length $K$, where $\alpha_j = 1$ and all other coefficients are zero.

| Length of LFSR ($k$) | Indices ($j$) |
|:---:|:---:|
| 2 | (0, 1) |
| 3 | (0, 1) |
| 4 | (0, 1) |
| 5 | (0, 2) |
| 6 | (0, 1) |
| 7 | (0, 1) |
| 8 | (0, 2, 3, 4) |
| 9 | (0, 4) |
| 10 | (0, 3) |
| 11 | (0, 2) |
| 12 | (0, 1, 2, 8) |
| 13 | (0, 1, 2, 5) |
| 14 | (0, 1, 2, 12) |
| 15 | (0, 1) |
| 16 | (0, 1, 3, 12) |
| 17 | (0, 3) |
| 18 | (0, 7) |
| 19 | (0, 1, 2, 5) |
| 20 | (0, 3) |
| 21 | (0, 2) |
| 22 | (0, 1) |
| 23 | (0, 5) |
| 24 | (0, 1, 2, 7) |

### A.2  LFSR STATE SEQUENCE ILLUSTRATION

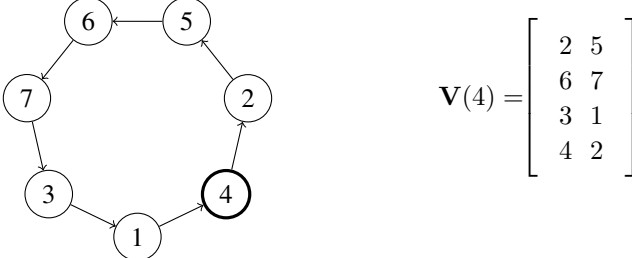

Figure 4: Illustration of the state sequence for a $K = 3$ LFSR with all possible states with the feedback polynomial defined in Table A.1. The matrix $\mathbf{V}(4)$ is generated from these states. The seed state $s = 4$ is highlighted with a thick circle.

## A.3 LFSR SEQUENCE GENERATION

This section explains how pseudo-random sequences are generated for our weight compression method using LFSRs which efficiently produce pseudo-random binary sequences, enabling on-the-fly generation. A key function generates a sequence of states by taking inputs such as a seed, the number of bits $(K)$ in the LFSR, the sequence length, and the positions of the taps. These taps, which determine which bits are XORed to produce the feedback bit, are chosen to ensure maximal length and are summarized in Table 6.

**Process Overview.** Starting with an initial state (seed), the LFSR computes the next state iteratively by:

Shifting all bits in the current state to the right by one position. XORing the bits at the specified tap positions to calculate the feedback bit, which is shifted into the leftmost position. This process is repeated for the desired sequence length, with the generated states stored in an array for downstream use.

The pseudocode for this process is provided in Algorithm 1.

---

**Algorithm 1** LFSR Sequence Generation

---

**Require:** $K$: Number of bits in the LFSR
       $seed$: Initial state of the LFSR
       $length$: Length of the sequence to generate
       $taps$: List of tap positions affecting feedback {Extracted from Table 6 for SeedLM}
1: Set $current\_state \leftarrow seed$.
2: **for** $i = 0$ to $length - 1$ **do**
3:    Initialize $b_k \leftarrow 0$. {$b_k$ is the new feedback bit}
4:    **for** each $tap$ in $taps$ **do**
5:       Update $b_k \leftarrow b_k \oplus ((current\_state \gg tap)\&1)$.
6:    **end for**
7:    Update $current\_state \leftarrow (current\_state \gg 1) \,|\, (b_k \ll (k - 1))$.
8:    Assign $results[i] \leftarrow current\_state$.
9: **end for**
10: **return** $results$

---

## A.4 Reconstruction Process in SeedLM

The reconstruction process of a weight matrix in SeedLM transforms compressed data back into its approximate original form using a combination of pseudo-random sequence generation and compressed coefficients. Given a set of seeds $\{s_i\}_{i=1}^{\text{num\_blocks}}$ and coefficients $\{\mathbf{t}_i\}_{i=1}^{\text{num\_blocks}}$, where $\mathbf{t}_i \in \mathbb{R}^P$, Algorithm 2 is performed:

---

**Algorithm 2** Reconstruction Process in SeedLM

---

**Require:** $K$: Number of bits in the LFSR

$\qquad \{s_i\}_{i=1}^{\text{num\_blocks}}$: Set of seeds

$\qquad \{\mathbf{t}_i\}_{i=1}^{\text{num\_blocks}}$: Set of coefficients, where $\mathbf{t}_i \in \mathbb{R}^P$

$\qquad C$: Block size

$\qquad P$: Latent dimension

1: Initialize $length \leftarrow 2^K - 1$.
2: Generate all LFSR states from 1 to $length$, store as $states$.
3: Initialize $blocks \leftarrow []$.
4: **for** $i = 1$ to num_blocks **do**
5: $\quad$ Compute $start \leftarrow s_i \% length$. {% denotes the remainder operation}
6: $\quad$ Extract a slice of $states$ starting at $start$ and containing $C.P$ elements. If the slice exceeds $length$, cycle through $states$.
7: $\quad$ Rearrange the slice into a matrix $\mathbf{U}(s_i) \in \mathbb{R}^{C \times P}$, following a row-wise order.
8: $\quad$ Compute $block \leftarrow \mathbf{U}(s_i) \times \mathbf{t}_i$ (matrix-vector multiplication).
9: $\quad$ Append $block$ to $blocks$.
10: **end for**
11: Rearrange $blocks$ to reconstruct the final weight matrix $output$.
12: **return** $output$

---

## A.5 SeedLM Algorithm

Algorithm 3 summarizes SeedLM's method, which operates in parallel across all weight blocks to identify the seeds and coefficients used to reconstruct weights. To enhance computational efficiency, we precompute and cache the pseudo-inverse matrices for all seeds and execute steps 2–5 concurrently across all blocks. Additionally, the inner loop can be parallelized. For the selected values of $C$, $P$, and $K$ used in Table 1, the pseudo-inverses require at most 6.3MB of memory, which is negligible.

---

**Algorithm 3** Seed and Coefficient Selection for a Weight Block

---

**Require:** $\mathbf{w} \in \mathbb{R}^C$, $\{\mathbf{U}(j)^\dagger\}_{j=1}^{2^K-1}$

1: $\hat{s}^* \leftarrow$ null, $\hat{\mathbf{t}}^* \leftarrow$ null
2: min_norm $\leftarrow \infty$
3: **for** $j = 1$ **to** $2^K - 1$ **do**
4: $\quad \mathbf{t} \leftarrow q(\mathbf{U}(j)^\dagger \mathbf{w})$, where $q(\cdot)$ quantizes its arguments to the set $\mathcal{Q}$
5: $\quad$ norm $\leftarrow \|\mathbf{w} - \mathbf{U}(j) \cdot \mathbf{t}\|$
6: $\quad$ **if** norm < min_norm **then**
7: $\qquad$ min_norm $\leftarrow$ norm
8: $\qquad \hat{s}^* \leftarrow s$
9: $\qquad \hat{\mathbf{t}}^* \leftarrow \mathbf{t}$
10: $\quad$ **end if**
11: **end for**
12: **return** $s^*$, and $\mathbf{t}^*$

---

## A.6 ABLATION STUDY ON 4-BIT COMPRESSION

We extend Table 2 to include the well-known GPTQ from (Frantar et al., 2022) for 4-bit weight compression. We compare the perplexity of various 4-bit weight compression techniques on WikiText-2 using 166 test windows, each containing 2048 tokens. As shown in the results, SeedLM outperforms other methods while requiring no calibration data or fine-tuning.

Table 7: WikiText-2 perplexity results for 4-bit representations of Llama 2 and Llama 3 models with a sequence length of 2048. The notation x-yB refers to the Llama x model with yB parameters (e.g., 2-7B means Llama 2 with 7 billion parameters). Perplexity values above 100 are shown as inf. The best perplexity values are highlighted.

| Method | Bits | 2-7B | 2-13B | 2-70B | 3-8B | 3-70B |
|---|---|---|---|---|---|---|
| Baseline | 16 | 5.5 | 4.9 | 3.3 | 6.1 | 2.9 |
| SeedLM | 4 | 5.7 | 5.1 | 3.5 | 7.0 | 3.8 |
| OmniQuant | 4 | 6.1 | 5.2 | 3.7 | inf | inf |
| AWQ | 4 | 5.8 | 5.1 | 3.5 | 7.1 | 4.7 |
| GPTQ | 4 | 8.3 | 5.7 | 3.9 | 8.2 | 6.7 |

