# OpenReview forum: "SeedLM: Compressing LLM Weights into Seeds of Pseudo-Random Generators"
_ICLR.cc/2025/Conference — ICLR 2025 Poster_

### Official Review · Reviewer_GgWo · 2024-10-27

**Soundness:** 3
**Presentation:** 2
**Contribution:** 2
**Rating:** 6
**Confidence:** 3

**Summary:**

The paper proposes SeedLM, a novel method for compressing large language model (LLM) weights by encoding them into seeds of pseudo-random generators.  SeedLM is a data-free, post-training compression method that achieves competitive accuracy at 3/4-bit compression context, as demonstrated on models like Llama 2 and Llama 3. The technique utilizes Linear Feedback Shift Registers (LFSRs) to generate pseudo-random matrices that reconstruct weight blocks during inference. Hardware implementation on an FPGA further supports the potential of SeedLM.

**Strengths:**

1. Weight compression using pseudo-random generator seeds is a novel-sounding technique. It enables significant compression while maintaining high accuracy.
2. Unlike many state-of-the-art compression methods, the proposed method does not require calibration data, reducing the need for correction data acquisition and potentially further reducing the quantization offset problem caused by the calibration data distribution.
3. The authors validated the computational characteristics and efficiency of their proposed algorithm using FPGA, and the FPGA implementation verifies the SeedLM in some ideal hardware-constrained environments.

**Weaknesses:**

1. The author compared the AWQ, Omniquant, and QuIP# methods. However, Omniquant and QuIP# were primarily designed for ultra-low bit-width quantization compression, such as 2-bit, but the author only compared the performance of 3/4-bit and did not show the quantization results of 2-bit. In the field of LLM quantization, SOTA methods specifically designed for 4/3-bit, such as GPTQ[1], were not included in the comparison. This makes the results unconvincing.

2. The author mentions in Section 4.1, lines 356-358, that to ensure a fair comparison with QuIP# and Omniquant, no fine-tuning was performed on them. This is a fair comparison for QuIP#, which combines codebook and fine-tuning of pre-trained parameters to improve performance. However, Omniquant does not fine-tune any pre-trained parameters, instead using block-wise gradient propagation to update the quantizer parameters, including the scaling factor and zero factor. By not using this technique in the comparison, the author is essentially not using the Omniquant method, but rather a basic statistical quantization. And, AWQ also uses calibration data to pre-compute the scaling parameters. This comparison is unfair and may cause confusion for readers. Additionally, the data in Table 2 is different from what is reported in the AWQ paper, and the author should provide an explanation for this discrepancy.

3. AWQ can quickly determine the scaling of weights and perform quantization through calibration, and the compression time for a 7B model is only a few minutes. However, the LFSR technique proposed in the paper involves matrix decomposition and optimization approximation, and the efficiency of this compression process for extremely large-scale LLMs is lacking in discussion and comparison.

4. The overall writing of the article is not clear in some details. In Eq (1), I can infer that the compression matrices for LFSR are **U** and **t**, but the details of how the input activation **X** is computed with the LFSR-compressed weights during the actual inference decoding stage are not discussed in detail. The article should provide more information on how the LFSR-compressed weights are used in the inference process and how they differ from other quantization methods. This would help to clarify the advantages and characteristics of the LFSR-based method.





[1] GPTQ: Accurate Post-Training Quantization for Generative Pre-trained Transformers.

**Questions:**

Please refer to the weaknesses items.

---

> ### Author Response · Authors · 2024-11-19
> **Response to Reviewer GgWo (Part 1)**
>
> Thank you for your thoughtful and detailed feedback. We greatly appreciate your recognition of the innovative aspects of SeedLM, especially its data-free approach and the use of pseudo-random generator seeds for efficient weight compression. We're pleased to hear that you found the FPGA implementation and its potential applications in hardware-constrained environments valuable. Eliminating the need for calibration data would indeed simplify the compression process, reduce quantization offset issues, and can maintain high accuracy while achieving good compression ratios.
>
> We appreciate your feedback and will carefully address the areas you highlighted to further improve the work. Below, we provide our responses to your specific points:
>
> > The author compared the AWQ, Omniquant, and QuIP# methods. However, Omniquant and QuIP# were primarily designed for ultra-low bit-width quantization compression, such as 2-bit, but the author only compared the performance of 3/4-bit and did not show the quantization results of 2-bit. In the field of LLM quantization, SOTA methods specifically designed for 4/3-bit, such as GPTQ[1], were not included in the comparison. This makes the results unconvincing.
>
> Thank you for pointing this out. SeedLM primarily focuses on 3- and 4-bit configurations, and we will make this clearer in the paper by adding the following to the discussion:
>
> "SeedLM primarily focused on 3-bit and 4-bit compression, applying the same design choices across all blocks for these levels. However, we believe that pushing below the 3-bit level might require different configurations tailored to each block to manage compression noise effectively or leveraging data for additional fine-tuning to enhance results. We leave these directions for future work."
>
> We appreciate you highlighting GPTQ as a strong method for 3/4-bit quantization. Initially, we did not include it in our comparisons because the baselines we selected were reported in the literature as superior or comparable to GPTQ. However, we recognize the importance of including GPTQ in our comparisons. Below, please find the perplexity results obtained using the same WikiText-2 setting as in Table 2 for 4-bit quantization. These results show that SeedLM consistently outperforms GPTQ in 4-bit configurations. For 3-bit quantization, GPTQ results appear significantly lower; for instance, for LLama2-13B in 3-bit, GPTQ yields a perplexity of 9.5 compared to SeedLM's 5.8. We will include these findings in the ablation section of the paper's appendix for completeness.
>
> | **Method**    | **Bits** | **2-7B** | **2-13B** | **2-70B** | **3-8B** | **3-70B** |
> |---------------|----------|----------|-----------|-----------|----------|-----------|
> | Baseline      | 16       | 5.5      | 4.9       | 3.3       | 6.1      | 2.9       |
> | SeedLM        | 4        | 5.7      | 5.1       | 3.5       | 7.0      | 3.8       |
> | OmniQuant     | 4        | 6.1      | 5.2       | 3.7       | inf      | inf       |
> | AWQ           | 4        | 5.8      | 5.1       | 3.5       | 7.1      | 4.7       |
> | QuIP#         | 4        | 6.5      | 5.3       | OOM       | 7.6      | OOM       |
> | GPTQ          | 4        | 8.3      | 5.7       | 3.9       | 8.2      | 6.7       |

---

> ### Author Response · Authors · 2024-11-19
> **Response to Reviewer GgWo (Part 2)**
>
> > The author mentions in Section 4.1, lines 356-358, that to ensure a fair comparison with QuIP# and Omniquant, no fine-tuning was performed on them. This is a fair comparison for QuIP#, which combines codebook and fine-tuning of pre-trained parameters to improve performance. However, Omniquant does not fine-tune any pre-trained parameters, instead using block-wise gradient propagation to update the quantizer parameters, including the scaling factor and zero factor. By not using this technique in the comparison, the author is essentially not using the Omniquant method, but rather a basic statistical quantization. And, AWQ also uses calibration data to pre-compute the scaling parameters. This comparison is unfair and may cause confusion for readers. Additionally, the data in Table 2 is different from what is reported in the AWQ paper, and the author should provide an explanation for this discrepancy.
>
> Thank you for your detailed observations. Regarding AWQ, the numbers reported in the AWQ paper are based on a group size of 128. As explained in our paper, we use 3- or 4-bit integers with channel-wise scaling to ensure that the required bits per element remain within the allocated 3 or 4 bits. A group size of 128, as used in AWQ, introduces an additional quantization overhead of approximately 0.25 bits per parameter. Consequently, the perplexity results for AWQ in Table 2 and the evaluation accuracies in Table 3 are based solely on channel-wise scaling. In this case, the AWQ codebase defaults to using the entire model channel width as the group size, resulting in negligible quantization overhead compared to the 0.25-bit overhead per parameter with a group size of 128. We will clarify this distinction further in the paper. Your point regarding OmniQuant is valid, and we will address it in the paper to ensure readers fully understand the differences in methodology. That said, it is important to highlight that SeedLM remains the only data-free approach in the comparison, achieving competitive or superior results. We appreciate your feedback and will incorporate these clarifications to enhance the paper.
>
> > AWQ can quickly determine the scaling of weights and perform quantization through calibration, and the compression time for a 7B model is only a few minutes. However, the LFSR technique proposed in the paper involves matrix decomposition and optimization approximation, and the efficiency of this compression process for extremely large-scale LLMs is lacking in discussion and comparison.
>
> Thank you for raising this important point. In the paper, we highlight that the heuristics are applied across all weight blocks in parallel to identify the seeds and coefficients that minimize reconstruction error, aiming to enhance computational efficiency. To achieve this, we precompute and cache the pseudo-inverse matrices for all seeds and perform the heuristics in parallel across all blocks.
>
> The seed search process is inherently parallelizable, as it can be conducted independently across both seeds and blocks, allowing efficient scaling for large models. While compression time varies depending on configuration parameters such as seed bit-width, block size, and latent dimension, the use of these parallelization strategies enables significant efficiency gains. For reference, we were able to perform all compression tasks, including up to a 70B-parameter model, using a single 80GB A100, with runtimes ranging from minutes to hours depending on the model size and configurations. We will elaborate on these details further in the paper.

---

> ### Author Response · Authors · 2024-11-19
> **Response to Reviewer GgWo (Part 3)**
>
> > The overall writing of the article is not clear in some details. In Eq (1), I can infer that the compression matrices for LFSR are U and t, but the details of how the input activation X is computed with the LFSR-compressed weights during the actual inference decoding stage are not discussed in detail. The article should provide more information on how the LFSR-compressed weights are used in the inference process and how they differ from other quantization methods. This would help to clarify the advantages and characteristics of the LFSR-based method.
>
> Thank you for your feedback. This approach differs from traditional quantization methods by leveraging a pseudo-random generator to dynamically reconstruct weights during inference. This design specifically targets speeding up memory-bound tasks, as highlighted in the paper. In response to your feedback, we will add the following explanation to the paper before Equation 1 to clarify the inference process:
>
> "During inference, $s$ and $\mathbf{t}$, which require fewer bits when considering all the bits in a block compared to the original weights, are retrieved from memory. This reduces the memory footprint and accelerates memory-bound generation tasks. The weights are reconstructed on-the-fly using $s$ and $\mathbf{t}$, and these reconstructed weights are then used to compute intermediate activations and, ultimately, the outputs."
>
> We appreciate your valuable feedback and hope our responses have addressed your concerns, warranting a re-evaluation of the score. We welcome further discussion of any remaining questions during the discussion period. Thank you for your time and consideration.

---

> > ### Comment · Reviewer_GgWo · 2024-11-23
> > **Reply to authors**
> >
> > Hi authors,
> >
> > Thank you for your detailed rebuttal. For the quantization process and time cost compared to Omniquant or AWQ, I would like to see an experiment table fairly comparing the efficiency during quantization (same model size, time for quantization). And for the computation detail and inner process of LFSR, I really hope to see detailed quantization and dequantization pseudocode, although we currently can't deploy and test your running code. This would be crucial for me to improve my rate. Thank you very much for your efforts.
> >
> > Good luck!

---

> > > ### Author Response · Authors · 2024-11-27
> > > **Response to Reviewer GgWo (Part 4)**
> > >
> > > Dear Reviewer,
> > >
> > > Thank you for your feedback and your willingness to consider raising your score. We appreciate your constructive comments on the efficiency of the quantization process and the computational details and inner workings of the LFSR. We have addressed these points in detail below.
> > >
> > > **Efficiency During Quantization**
> > >
> > > As you suggested, we compare SeedLM with AWQ for quantization time and FLOPs, as AWQ efficiently utilizes activations among the data-adapted approaches. For a LLaMA2 7B model, both methods can take less than 20 minutes on the same A100 GPU. From our experience, SeedLM can be faster for the implementations we used compared to the AWQ repository for achieving similar results on LLaMA2 7B. The speed difference is also perceivable in FLOPs, where SeedLM requires fewer FLOPs for compression. Below, we outline the details of the FLOPs.
> > >
> > > AWQ involves finding $\alpha \in [0,1]$ via minimizing the expression using grid search:
> > >
> > > $\lVert Q(\mathbf{W} \cdot \text{Diag}(\mathbf{s}^\alpha_\mathbf{X})) \cdot \text{Diag}(\mathbf{s}_\mathbf{X}^\alpha)^{-1} \mathbf{X} - \mathbf{W} \cdot \mathbf{X} \rVert,$
> > >
> > > where $\mathbf{W} \in \mathbb{R}^{d_1 \times d_2}$, $\mathbf{X} \in \mathbb{R}^{d_2 \times L}$ (with $L$ being the sequence length), $\mathbf{s}_\mathbf{X}$ is the average magnitude of activation (per-channel), and $\alpha$ is iterated over 20 options (the grid size used to optimize $\alpha$, see also Section 5.1 in AWQ paper).
> > >
> > > To get a theoretical sense of quantization time, we compute the FLOPs involved:
> > >
> > >
> > > 1. Constructing $\text{Diag}(\mathbf{s}_\mathbf{X}^\alpha)$: Costs $\mathcal{O}(d_2)$ for each $\alpha$.
> > >
> > > 2. Computing $\mathbf{W} \cdot \text{Diag}(\mathbf{s}_\mathbf{X}^\alpha)$: Costs $\mathcal{O}(d_1 \cdot d_2)$.
> > >
> > > 1. Applying $Q$: Denote its cost as $\mathcal{O}(q)$.
> > > 2. Computing $\text{Diag}(\mathbf{s}_\mathbf{X}^\alpha)^{-1} \cdot \mathbf{X}$: Costs $\mathcal{O}(d_2 \cdot L)$.
> > >
> > > 5. Matrix multiplication $Q(\mathbf{W} \cdot \text{Diag}(\mathbf{s}^\alpha_\mathbf{X})) \cdot \text{Diag}(\mathbf{s}_\mathbf{X}^\alpha)^{-1} \mathbf{X}$: Costs $\mathcal{O}(2 \cdot d_1 \cdot d_2 \cdot L)$ because it involves both multiplications and additions.
> > >
> > > 1. Computing $\mathbf{W} \cdot \mathbf{X}$: Costs $\mathcal{O}(2 \cdot d_1 \cdot d_2 \cdot L)$.
> > >
> > > The overall FLOPs are primarily dominated by steps 5 and 6, as the norm computation and subtraction also incur a cost of $\mathcal{O}(d_1 \cdot L)$. Therefore, the total computational cost per $\alpha$ and batch is $\mathcal{O}(4 \cdot d_1 \cdot d_2 \cdot L).$
> > >
> > > Given that $\alpha$ is iterated over 20 grid points and this computation is applied across $B$ batches, the total FLOPs become $\mathcal{O}(20 \cdot B \cdot (4 \cdot d_1 \cdot d_2 \cdot L)).$
> > >
> > > For the LLMs in question, the minimum $L$ used by AWQ is 512, and the minimum number of samples considered is 128. So the overall FLOPs for AWQ quantization is greater than $\mathcal{O}(20 \cdot 128 \cdot (4 \cdot d_1 \cdot d_2 \cdot 512)) > \mathcal{O}(2^{22} \cdot d_1 \cdot d_2).$
> > >
> > > On the other hand, SeedLM does not require $X$ (activations). Instead, it divides $\mathbf{W}$ into $\frac{d_1 d_2}{C}$ blocks (where $C$ is the block size) and estimates each block by solving the optimization problem:
> > >
> > > $\underset{s, \mathbf{t}}{\text{minimize}} \quad \lVert \mathbf{w} - \mathbf{U}(s) Q(\mathbf{t}) \rVert_2^2, \quad \text{subject to:} \quad s \in \{1, \dots, 2^K - 1\},$
> > >
> > > where $\mathbf{w} \in \mathbb{R}^{C \times P}$ and $\mathbf{t} \in \mathbb{R}^{P \times 1}$. Similarly, calculating the norm for each seed is dominated by the cost $\mathcal{O}(2 \cdot C \cdot P)$, which must be performed for $\frac{d_1 d_2}{C}$ blocks and $2^K$ seeds, leading to a total cost of $\mathcal{O}(2^K \cdot 2 \cdot C \cdot P \cdot \frac{d_1 d_2}{C}) = \mathcal{O}(2^{K+1} \cdot P \cdot d_1 \cdot d_2).$
> > >
> > > For SeedLM's 3-bit and 4-bit configurations, $K = 16$ and $P$ is at most 4, resulting in a computational cost of $\mathcal{O}(2^{19} \cdot d_1 \cdot d_2)$ which is lower than AWQ's FLOPs lower bound of $\mathcal{O}(2^{22} \cdot d_1 \cdot d_2)$.
> > >
> > > This provides insights into why SeedLM can be more efficient and faster in compressing compared to data-adapted methods like AWQ, as it avoids the need for activations and significantly reduces computational overhead.

---

> > > ### Author Response · Authors · 2024-11-27
> > > **Response to Reviewer GgWo (Part 5)**
> > >
> > > **Detailed Quantization and Dequantization Pseudocode**
> > >
> > > We understand the importance of transparency in explaining our methods. In response to your feedback, we’ve uploaded a revised version of the paper. The newly added Appendices A.2 to A.4, along with Algorithms 1-3, offer a clearer view of the inner workings of the LFSR, as well as the compression and decompression of the weights, complementing the explanations in the main body of the paper.
> > >
> > > ---
> > >
> > > We hope this detailed response addresses your concerns. Thank you again for your constructive feedback and support. We also want to assure you that we will incorporate the insights from this discussion into the camera-ready version. We hope that this warrants a reevaluation of the score. Please let us know if you have any comments, as we welcome further discussion of any remaining questions during the discussion period.

---

> > > > ### Comment · Reviewer_GgWo · 2024-11-27
> > > > **Thank you for your efforts, increasing to 6**
> > > >
> > > > Thank you for your informative comparison of quantization time between your method and AWQ. I now understand better. I am raising my score to 6.
> > > >
> > > > Regarding Algorithm 3, is the coefficient selection done offline? While it can be computed in parallel, would it pose a significant burden if handled online?

---

> > > > > ### Author Response · Authors · 2024-11-27
> > > > > **Response to Reviewer GgWo (Part 6)**
> > > > >
> > > > > We’re glad that our response was helpful, and we greatly appreciate you taking the time to reassess our work. Thank you for your positive assessment of our work and for raising the score.
> > > > >
> > > > > To your question regarding online versus offline processing, SeedLM is an offline algorithm in that the coefficient selection and compression processes are performed entirely offline by accessing only the weights. This process can be executed in parallel across weight blocks. Once the compression is completed, the compressed form (seeds and coefficients) is used during runtime to reconstruct the weights on the fly using Algorithm 2. Accessing this compressed form during runtime reduces memory access, and can speed up memory-bound tasks like autoregressive generation.
> > > > >
> > > > > We acknowledge that online compression could add a burden on top of already having to access the full-precision weights. To avoid this, we suggest compressing the model first before serving the model or using it during runtime.
> > > > >
> > > > > Thanks again and please let us know if you have further questions!

---

### Official Review · Reviewer_WtPj · 2024-10-28

**Soundness:** 3
**Presentation:** 3
**Contribution:** 3
**Rating:** 6
**Confidence:** 5

**Summary:**

The paper introduces SeedLM, a novel post-training compression technique leveraging seeds of pseudo-random generators, specifically using Linear Feedback Shift Registers (LFSRs), to encode and compress weights in large language models (LLMs). SeedLM achieves memory efficiency by encoding model weights into compact seeds, reconstructing weights at runtime, and minimizing memory access during inference. Experimental results show its performance in zero-shot tasks, achieving competitive accuracy retention at 3- and 4-bit quantization levels, especially with the LLaMA 3 model, and demonstrating improved latency and throughput in FPGA implementations.

**Strengths:**

1- **Innovative Use of Arbitrary Data Formats**: The adoption of arbitrary data formats with shared exponents is a commendable design choice, enhancing the flexibility of SeedLM's quantization approach.

2- **Efficient FPGA Implementation**: Proposing an efficient FPGA implementation demonstrates the hardware viability of SeedLM and highlights potential real-world deployment in resource-constrained environments.

3- **Data-Free Compression**: SeedLM operates without calibration data, which differentiates it from many other compression methods that rely on data for fine-tuning and accuracy adjustments.

4- **Memory Efficiency**: By using seeds for reconstructing weights, SeedLM minimizes memory footprint, crucial for memory-bound applications like LLM inference, and shows considerable speedup, particularly on FPGAs.

**Weaknesses:**

1- **Absence of GPTQ Comparison**: The paper does not provide a comparison with GPTQ, a commonly used quantization baseline, which is a notable omission given GPTQ's relevance to LLM compression.

2- **Inference Efficiency Assumptions**: While the paper mentions using the latest repositories, many of these codebases likely store compressed weights in full precision during inference, leading to potential memory inefficiencies.

3- **GPU Implementation Challenges**: Although FPGA implementation is shown, the challenges of porting SeedLM to GPUs are unaddressed. Issues like increased kernel launches for memory-bound tasks and limited support for LFSRs in recent GPU hardware could impact performance.

4- **Optimization Overhead for Parameter Selection**: The process of determining optimal parameters for each weight block, such as the seeds, coefficients, and latent dimensions, may introduce significant overhead.

**Questions:**

1- **Comparison with GPTQ** : Why was GPTQ omitted from the comparisons? Can you clarify its potential impact on results, given that it’s a standard benchmark in LLM compression?

2- **Compression Limitations for LLaMA 3**: Many methods struggle to compress LLaMA 3 effectively. Do the authors have insights on why this model, in particular, is challenging to compress?

3- **Full Precision Storage Impact**: Have the authors tested SeedLM with full precision storage during inference? If so, could this be the cause of out-of-memory issues observed with some baseline methods?

4- **GPU Implementation Challenges**: Could the authors comment on the challenges of implementing SeedLM on GPUs, specifically the support for LFSR in recent GPUs and whether additional kernel launches could reduce performance for memory-bound applications on NVIDIA hardware?

5- **Parameter Optimization Overhead**: What is the computational cost of finding optimal seeds and coefficients? Could the process be streamlined for faster deployment?

6- **Sensitivity in Compressing LLaMA 3**: Many quantization techniques, appear to struggle with LLaMA 3, indicating potential limitations in compressing this model type. What features does this family of models have that makes it difficult for other methods to quantize, and why does SeedLM perform well in this case?

---

> ### Author Response · Authors · 2024-11-20
> **Response to Reviewer WtPj (Part 1)**
>
> Thank you for your thoughtful and detailed feedback. We’re delighted that you appreciated SeedLM’s approach, especially its innovative use of arbitrary data formats with shared exponents and its focus on a hardware-friendly FPGA implementation. We’re pleased to see recognition of SeedLM’s data-free nature, which distinguishes it from methods relying on calibration data or fine-tuning. Additionally, we appreciate your acknowledgment of the memory efficiency and speedup advantages.
> We deeply value your feedback and will carefully address the areas you highlighted to further enhance the work. Below, we provide detailed responses to your specific points:
>
> > **Weakness 1:** Absence of GPTQ Comparison: The paper does not provide a comparison with GPTQ, a commonly used quantization baseline, which is a notable omission given GPTQ's relevance to LLM compression.
>
> > **Question 1:** Comparison with GPTQ : Why was GPTQ omitted from the comparisons? Can you clarify its potential impact on results, given that it’s a standard benchmark in LLM compression?
>
> We acknowledge GPTQ as a strong compression method, and the reason we initially excluded it from our comparisons was that the baselines we selected were reported in the literature as either superior or comparable to GPTQ. However, we recognize the importance of including GPTQ in our evaluations. Below, please find the perplexity results obtained using the same WikiText-2 setting as in Table 2 for 4-bit quantization. These results show that SeedLM consistently outperforms GPTQ in 4-bit configurations. For 3-bit quantization, GPTQ results appear significantly lower; for instance, for LLama2-13B in 3-bit, GPTQ yields a perplexity of 9.5 compared to SeedLM's 5.8. We will include these findings in the ablation section of the paper's appendix for completeness.
>
> | Method    | Bits | 2-7B | 2-13B | 2-70B | 3-8B | 3-70B |
> |---------------|----------|----------|-----------|-----------|----------|-----------|
> | Baseline      | 16       | 5.5      | 4.9       | 3.3       | 6.1      | 2.9       |
> | SeedLM        | 4        | 5.7      | 5.1       | 3.5       | 7.0      | 3.8       |
> | OmniQuant     | 4        | 6.1      | 5.2       | 3.7       | inf      | inf       |
> | AWQ           | 4        | 5.8      | 5.1       | 3.5       | 7.1      | 4.7       |
> | QuIP#         | 4        | 6.5      | 5.3       | OOM       | 7.6      | OOM       |
> | GPTQ          | 4        | 8.3      | 5.7       | 3.9       | 8.2      | 6.7       |
>
> > **Weakness 2:** Inference Efficiency Assumptions: While the paper mentions using the latest repositories, many of these codebases likely store compressed weights in full precision during inference, leading to potential memory inefficiencies.
>
> > **Question 3:** Full Precision Storage Impact: Have the authors tested SeedLM with full precision storage during inference? If so, could this be the cause of out-of-memory issues observed with some baseline methods?
>
> Thank you for your observation. SeedLM is specifically designed to avoid storing weights in full precision during inference by using a custom decoding mechanism that reconstructs weights on-the-fly during inference. This approach minimizes memory usage, speeding up memory-bound tasks without the need for full-precision storage. The out-of-memory issue observed with Quip# is due to the potential inefficiencies in the Quip# repo. We’re working with the Quip# team to address this issue and will ensure the manuscript is updated accordingly.

---

> ### Author Response · Authors · 2024-11-20
> **Response to Reviewer WtPj (Part 2)**
>
> > **Weakness 3:** GPU Implementation Challenges: Although FPGA implementation is shown, the challenges of porting SeedLM to GPUs are unaddressed. Issues like increased kernel launches for memory-bound tasks and limited support for LFSRs in recent GPU hardware could impact performance.
>
> > **Question 4:** GPU Implementation Challenges: Could the authors comment on the challenges of implementing SeedLM on GPUs, specifically the support for LFSR in recent GPUs and whether additional kernel launches could reduce performance for memory-bound applications on NVIDIA hardware?
>
> Thank you for highlighting this point. Implementing SeedLM on GPUs does present challenges, such as increased kernel launches for memory-bound tasks and the limited support for LFSRs in current GPU hardware. These factors can affect performance, as GPUs are less optimized for the on-the-fly weight reconstruction approach used in SeedLM. Additionally, the data type for the latent vector may not be natively supported, adding to the complexity. That said, it’s important to note that hardware capabilities—across various platforms, not just GPUs—are continually evolving. LFSR is a particularly hardware-friendly method, well-suited to low-area, low-energy, and high-throughput implementations. One of the key aims of this paper is to demonstrate the practicality of SeedLM’s approach, including LFSR-based techniques, and to provide evidence that could motivate the development of new hardware features to better support such methods. While we focused on FPGA for its current advantages, we will add the following discussion to the paper to address broader possibilities and future hardware compatibility:
>
> “as hardware continues to evolve, SeedLM's applicability could expand, with its LFSR-based design offering low-area, low-energy, and high-throughput benefits. This work highlights the practicality of such techniques, aiming to inspire hardware innovations that support broader device compatibility, including GPUs.”
>
> > **Weakness 4:** Optimization Overhead for Parameter Selection: The process of determining optimal parameters for each weight block, such as the seeds, coefficients, and latent dimensions, may introduce significant overhead.
>
> > **Question 5:** Parameter Optimization Overhead: What is the computational cost of finding optimal seeds and coefficients? Could the process be streamlined for faster deployment?
>
> Thank you for raising this important point. In the paper, we highlight that the heuristics are applied across all weight blocks in parallel to identify the seeds and coefficients that minimize reconstruction error, aiming to enhance computational efficiency. To achieve this, we precompute and cache the pseudo-inverse matrices for all seeds and perform the heuristics in parallel across all blocks.
>
> The seed search process is inherently parallelizable, as it can be conducted independently across both seeds and blocks, allowing efficient scaling for large models. While compression time varies depending on configuration parameters such as seed bit-width, block size, and latent dimension, the use of these parallelization strategies enables significant efficiency gains. For reference, we were able to perform all compression tasks, including up to a 70B-parameter model, using a single 80GB A100, with runtimes ranging from minutes to hours depending on the model size and configurations. We will elaborate on these details further in the paper.

---

> ### Author Response · Authors · 2024-11-20
> **Response to Reviewer WtPj (Part 3)**
>
> > **Question 2:** Compression Limitations for LLaMA 3: Many methods struggle to compress LLaMA 3 effectively. Do the authors have insights on why this model, in particular, is challenging to compress?
>
> > **Question 6:** Sensitivity in Compressing LLaMA 3: Many quantization techniques, appear to struggle with LLaMA 3, indicating potential limitations in compressing this model type. What features does this family of models have that makes it difficult for other methods to quantize, and why does SeedLM perform well in this case?
>
> Thank you for this insightful observation. Compressing LLaMA 3 indeed presents unique challenges, likely due to its extensive training on a vast number of tokens. As mentioned in the paper, this is probably a result of its advanced architecture and significantly larger training dataset. The LLaMA 3 introduction in [Meta’s blog](https://ai.meta.com/blog/meta-llama-3/) states:
>
> “We made several new observations on scaling behavior during the development of Llama 3. For example, while the Chinchilla-optimal amount of training compute for an 8B parameter model corresponds to ~200B tokens, we found that model performance continues to improve even after the model is trained on two orders of magnitude more data. Both our 8B and 70B parameter models continued to improve log-linearly after we trained them on up to 15T tokens.”
>
> Models trained on such large datasets capture intricate patterns, which may make them more sensitive to compression techniques. This aligns with insights from the recent ICLR submission "[Scaling Laws for Precision](https://openreview.net/forum?id=wg1PCg3CUP)," which just came to our attention. The study observes that degradation from post-training quantization increases with the volume of training data, making models like LLaMA 3 more susceptible to performance loss during compression. We will incorporate this insight and citation into our paper to further clarify the compression challenges faced by highly trained models. As highlighted in Appendix G of this study, the authors speculate that this sensitivity may be arising from sharpness and hierarchical learning strategies developed during training. This suggests that LLaMA 3's sensitivity to compression may stem from its reliance on deeply hierarchical and sharply tuned features, making traditional data-reliant compression methods less effective. Data-dependent approaches likely require a substantial number of tokens to adapt effectively to such highly trained models, potentially defeating the purpose of efficient post-training compression. In this context, we speculate that SeedLM incurs a lower initial compression error when cast into its compressed form, keeping it closer to the model's final pretrained state. Its data-free approach appears particularly well-suited, as it can eliminate the need for additional adaptation data. SeedLM’s ability to succeed without such data highlights its robustness in handling the intricate structure of these advanced models.
>
> Thank you once again for your thoughtful feedback. We hope our responses have effectively addressed your concerns and merit a re-evaluation of the score. We welcome further discussion and are happy to address any additional questions during the discussion period. We appreciate your time and consideration.

---

> ### Comment · Reviewer_WtPj · 2024-11-22
>
> Dear Authors,
>
> I appreciate the detailed discussion to address my questions and concerns and look forward to seeing these clarifications and improvements incorporated into a revised version or the camera-ready manuscript. Additionally, I am eager to see how you address the memory-related challenges in comparison to related work. I maintain my positive assessment of your submission with a score of 6.
>
> Best regards,
>
> Reviewer WtPj

---

> > ### Author Response · Authors · 2024-11-23
> > **Response to Reviewer WtPj**
> >
> > Dear Reviewer,
> >
> > Thank you again for your thoughtful and constructive comments throughout this process. We are working with the Quip# team to address the issue you highlighted and will ensure that the camera-ready version incorporates these updates and reflects your feedback.
> >
> > We greatly appreciate your positive assessment of our work and truly value the opportunity to have engaged with you during this discussion period.

---

### Official Review · Reviewer_i822 · 2024-11-02

**Soundness:** 3
**Presentation:** 4
**Contribution:** 3
**Rating:** 6
**Confidence:** 3

**Summary:**

This is an interesting method of quantization, using pseudo-random generator to point to almost evenly distributed codebook items and fast adjustments.
The paper has high quality presentation with necessary formulas and diagrams
The paper has shortcomings in comparisons with the other methods, specifically it lacks finetuning upside analysis.

The paper may qualify for acceptance if the weaknesses are reasonably addressed in the review process.

**Strengths:**

Novel method to store and retrieve codes with pseudo-random number generator.
High quality presentation with necessary formulas and diagrams
Attention to implementation details in Performance Analysis section.

**Weaknesses:**

1) Comparison with other quantization methods is incomplete. Most striking shortcoming is lack of comparison with finetuned model which is what most current SOTA models use.

2) The paper dismisses comparison with strong methods like AQ, SPQR in desire to "avoid costly training". Yet these are quite good benchmarks to compare with, theu=y have reported figures, and to larg share of practitioners the extra training time (hours actually) could be acceptable.

3) some of the results in table 2 are not consistent with those published in the respective papers.
for instance, AWQ claims 3.41 perplexity  for L2-70, (https://arxiv.org/pdf/2306.00978, table 4) while you indicate 3.5 (Table 2)

**Questions:**

1) Please compare your method with SOTA methods which use finetuning after quantization.
2) What is your method performance when quantizing into 1.5-2.5 bits per weight? This area is where most quantization research progress is happening, your method may be competitive there too.
3) please address the OOM situation in Quip# L2-70 benchmarks (Table2). What is different in your setup vs the original one from https://arxiv.org/pdf/2307.13304? What have you tried to avoid OOM?
4) Add performance timing benchmarks vs other methods, this solidifies your claim of speed advantage.
5) How would speedup of the method change after post-quantization finetuning?
6) Unlike some other submission, this one has no implementation code. I wonder if a private code repository could be provided for review purposes?
7) Please provide comparison to AQ, SPQR, as these are strong and valid SOTA benchmarks

---

> ### Author Response · Authors · 2024-11-21
> **Response to Reviewer i822 (Part 1)**
>
> Thank you for your thoughtful and detailed feedback. We’re pleased that you found the novelty of our method, particularly the use of a pseudo-random generator for efficient code storage and retrieval, to be valuable. We’re also glad that the quality of our presentation, including the detailed formulas, diagrams, and the implementation focus in the Performance Analysis section, resonated with you. We are actively working to address the comments you raised, which we believe will further strengthen the paper. Below, we provide detailed responses to your specific points:
>
> > **Weakness 1:** Comparison with other quantization methods is incomplete. Most striking shortcoming is lack of comparison with finetuned model which is what most current SOTA models use.
>
> > **Question 1:** Please compare your method with SOTA methods which use finetuning after quantization.
>
> Thank you for raising this insightful point. While fine-tuning after quantization certainly has its merits, our work offers a complementary perspective with significant potential. SeedLM achieves state-of-the-art results in 3/4-bit quantization without requiring any data. Additionally, beyond the simplicity of a data-free approach, it may help mitigate compression offset issues, especially when the distribution of fine-tuning data is not plausible. That said, we acknowledge the importance of exploring fine-tuning strategies on top of SeedLM, which we have highlighted in the concluding remarks as a future direction. We are actively investigating effective fine-tuning strategies for SeedLM and agree with you that it has the potential to push the bit-rate even further down while maintaining near iso-accurate performance. While we couldn’t conclude these experiments in the rebuttal period, we will ensure that our findings are included in the final version of the paper, should it be accepted.
>
> > **Weakness 2:** The paper dismisses comparison with strong methods like AQ, SPQR in desire to "avoid costly training". Yet these are quite good benchmarks to compare with, theu=y have reported figures, and to larg share of practitioners the extra training time (hours actually) could be acceptable.
>
> > **Question 7:** Please provide comparison to AQ, SPQR, as these are strong and valid SOTA benchmarks
>
> Thank you for highlighting this point. These are indeed excellent benchmarks, and we agree that they represent strong and valid SOTA comparisons. We are actively working to obtain those numbers and will include them in the paper. Thank you for bringing these up.
>
> > **Weakness 3:** some of the results in table 2 are not consistent with those published in the respective papers. for instance, AWQ claims 3.41 perplexity for L2-70, (https://arxiv.org/pdf/2306.00978, table 4) while you indicate 3.5 (Table 2)
>
> Thank you for your detailed observations. Regarding AWQ, the numbers reported in the AWQ paper are based on a group size of 128. As explained in our paper, we use 3- or 4-bit integers with channel-wise scaling to ensure that the required bits per element remain within the allocated 3 or 4 bits. A group size of 128, as used in AWQ, introduces an additional quantization overhead of approximately 0.25 bits per parameter. Consequently, the perplexity results for AWQ in Table 2 and the evaluation accuracies in Table 3 are based solely on channel-wise scaling. In this case, the AWQ codebase defaults to using the entire model channel width as the group size, resulting in negligible quantization overhead compared to the 0.25-bit overhead per parameter with a group size of 128. We will clarify this distinction further in the paper.
>
> > **Question 2:** What is your method performance when quantizing into 1.5-2.5 bits per weight? This area is where most quantization research progress is happening, your method may be competitive there too.
>
> Thank you for pointing this out. SeedLM primarily focuses on 3- and 4-bit configurations, as going below 3 bits without data may require custom adjustments (e.g., seed bit-width, block size, latent dimension, and latent data type) to better address compression noise based on the weight values. Alternatively, it could involve leveraging data and applying fine-tuning on the models, adapters, or similar adjustments. We believe that achieving competitive results with SeedLM below 3 bits starts to depend on such techniques, and we will elaborate on these points in the paper as discussed.
>
> > **Question 3:** please address the OOM situation in Quip# L2-70 benchmarks (Table2). What is different in your setup vs the original one from https://arxiv.org/pdf/2307.13304? What have you tried to avoid OOM?
>
> Thank you for your observation. The out-of-memory issue observed with Quip# is due to the potential inefficiencies in the Quip# repo. We’re working with the Quip# team to address this issue and will ensure the manuscript is updated accordingly.

---

> > ### Author Response · Authors · 2024-11-21
> > **Response to Reviewer i822 (Part 3)**
> >
> > We also speculate that SeedLM may address weight distribution distortion in highly trained models. For example, a recent ICLR submission, "[Scaling Laws for Precision](https://openreview.net/forum?id=wg1PCg3CUP)," observes that degradation from post-training quantization increases with the volume of training data. This makes models like LLaMA 3 more susceptible to performance loss during compression, requiring a substantial number of tokens to adapt effectively to such highly trained models. In this context, SeedLM's data-free approach appears particularly well-suited for models like LLaMA 3, which has been trained on an extensive number of tokens, as noted in [Meta’s blog](https://ai.meta.com/blog/meta-llama-3/):
> >
> > “We made several new observations on scaling behavior during the development of Llama 3. For example, while the Chinchilla-optimal amount of training compute for an 8B parameter model corresponds to ~200B tokens, we found that model performance continues to improve even after the model is trained on two orders of magnitude more data. Both our 8B and 70B parameter models continued to improve log-linearly after we trained them on up to 15T tokens.”
> >
> > As highlighted in Appendix G of this "[Scaling Laws for Precision](https://openreview.net/forum?id=wg1PCg3CUP)", the authors speculate that this sensitivity may stem from sharpness and hierarchical learning strategies developed during training. Data-dependent approaches may require a substantial number of tokens to adapt effectively to such highly trained models, potentially undermining the purpose of efficient post-training compression.
> >
> > In contrast, we speculate that SeedLM incurs a lower initial compression error when cast into its compressed form, preserving a state closer to the model's final pretrained condition. Its data-free approach appears particularly advantageous, as it eliminates the need for additional adaptation data. We will incorporate this insight and citation into our paper. Thank you again.

---

> > ### Comment · Reviewer_i822 · 2024-11-22
> >
> > Dear Authors,
> > thank you for addressing my comments and concerns. I look forward to see the promised improvements reflected in the camera-ready version.
> > For comparison with current SOTA I suggest to pick the most relevant ones from this list: Quip#, AQLM, AWQ, GPTQ, SquezeLLM , QTIP. Some of them show better perplexity on Llama-2 than your method, would be unfair to omit that.
> > I share the concerns of Reviewer WtPj regarding GPU implementation. I encourage you to be more specific about obstacles in GPU implementation. This matters a lot for the industry adoption of a quantization method.
> > I retain my rating of 6.

---

> ### Author Response · Authors · 2024-11-21
> **Response to Reviewer i822 (Part 2)**
>
> > **Question 4:** Add performance timing benchmarks vs other methods, this solidifies your claim of speed advantage.
>
> Thank you for this suggestion. Each method we evaluated requires specific hardware adjustments to optimize performance, which makes direct comparisons challenging. In future work, we’ll explore ways to standardize hardware setups to facilitate clearer comparisons. For now, we chose bit width as a standardized metric because it is widely used in the literature and reflects the weight's average bits per element, which directly influences memory access and correlates with speedups in memory-bound tasks.
>
> > **Question 5:** How would speedup of the method change after post-quantization finetuning?
>
> Thank you for the question. Post-quantization finetuning would have a minimal impact on inference latency, as runtime is not heavily influenced by the specific learned weights. SeedLM’s speed advantage primarily arises from the efficient use of pseudo-randomly generated weights and reduced memory access, which remain consistent regardless of whether finetuning is applied.
>
> > **Question 6:** Unlike some other submission, this one has no implementation code. I wonder if a private code repository could be provided for review purposes?
>
> Unfortunately, due to our institution's policy, we’re unable to comply with the anonymized requirements outlined in the ICLR guidelines. We recognize the importance of reviewing both the code and the reported findings, and we’re committed to working with our institution to find ways to share the code publicly once the review is complete.
>
> Thank you for your valuable feedback. We hope our responses have addressed your concerns and will incorporate the changes in the paper as discussed. If you have any remaining questions, we’re happy to discuss them further during the review period. We appreciate your time and consideration.

---

> ### Author Response · Authors · 2024-11-23
> **Response to Reviewer i822**
>
> Dear Reviewer,
>
> Thank you for your thoughtful comments and for providing a helpful list of current SOTA methods to consider. We are working to include comparisons with some of these methods, as you suggested. We will also ensure that the camera-ready version includes a detailed discussion of the GPU implementation challenges, as requested. We understand the importance of this aspect for industry adoption and will address it accordingly.
>
> Thank you again for your constructive feedback and positive assessment of our work. We truly value having had the opportunity to engage with you during this discussion period.

---

### Meta-Review · Area_Chair_97xa · 2024-12-22

**Metareview:**

This paper presents SeedLM, a novel post-training compression method for LLMs that finds input seeds from model weights for pseudo-random generators, specifically using Linear Feedback Shift Registers (LFSRs). By reconstructing weights at runtime from compact seeds, SeedLM significantly reduces memory usage while maintaining competitive accuracy.

The method operates without requiring retraining or calibration data, achieving efficient compression at 3- and 4-bit quantization levels, as demonstrated on models such as LLAMAs. Experimental results highlight its strong zero-shot performance, and hardware implementations on FPGAs show its improved throughput.

While this paper makes meaningful contributions, it has a few notable weaknesses. Several reviewers pointed out the lack of comparisons with current state-of-the-art methods, insufficient details on the additional overhead introduced by SeedLM and how it compares to baseline approaches, missing GPU implementation and code details, and the absence of explanations regarding the handling of LLaMA models and OOM cases.

Despite these issues, they do not detract from the paper’s overall contributions. Therefore, I lean toward recommending acceptance.

**Additional Comments On Reviewer Discussion:**

During the rebuttal period, most reviewers initially leaned toward borderline rejection. Most concerns are about the lack of comparisons with SoTA methods, questions about the reliability of the reported results (e.g., inconsistencies with prior work), and insufficient methodological details that hindered a full understanding of the proposed approach. However, the authors provided clarifications and additional results during the rebuttal, which addressed many of these concerns. As a result, most reviewers were convinced by the responses and shifted their scores toward acceptance.

---

### Decision · Program_Chairs · 2025-01-22

Accept (Poster)